# Effect of Rootstock on the Volatile Profile of Mandarins

**DOI:** 10.3390/foods12081599

**Published:** 2023-04-10

**Authors:** María Ángeles Forner-Giner, Paola Sánchez-Bravo, Francisca Hernández, Amparo Primo-Capella, Marina Cano-Lamadrid, Pilar Legua

**Affiliations:** 1Instituto Valenciano de Investigaciones Agrarias (IVIA), 46113 Moncada, Spain; 2Centro de Investigación e Innovación Agroalimentaria y Agroambiental (CIAGRO-UMH), Ctra. Beniel, km 3.2, 03312 Orihuela, Spain; 3Laboratorio de Fitoquímica y Alimentos saludables (LabFAS), Departmento de Ciencia y Tecnología de Alimentos, Centro de Edafología y Biología Aplicada del Segura (CEBAS)-CSIC, University Campus-25, 30100 Murcia, Spain; 4Postharvest and Refrigeration Group, Department of Agronomical Engineering and Institute of Plant Bio-Technology, Universidad Politécnica de Cartagena, 30203 Murcia, Spain

**Keywords:** citrus fruits, clemenules, Gas Chromatography/Mass Spectrometry (GC-MS), hybrid, juice, volatile compounds

## Abstract

Mandarin production has increased in recent years, especially for fresh consumption, due to its ease of peeling, its aroma, and its content of bioactive compounds. In this sense, aromas play a fundamental role in the sensory quality of this fruit. The selection of the appropriate rootstock is crucial for the success of the crop and its quality. Therefore, the objective of this study was to identify the influence of 9 rootstocks (“Carrizo citrange”, “Swingle citrumelo CPB 4475”, “Macrophylla”, “Volkameriana”, “Forner-Alcaide 5”, “Forner-Alcaide V17”, “C-35”, “Forner-Alcaide 418”, and “Forner-Alcaide 517”) on the volatile composition of “Clemenules” mandarin. For this, the volatile compounds of mandarin juice were measured using headspace solid-phase micro-extraction in a gas chromatograph coupled to a mass spectrometer (GC-MS). Seventy-one volatile compounds were identified in the analyzed samples, with limonene being the main compound. The results obtained showed that the rootstock used in the cultivation of mandarins affects the volatile content of the juice, with “Carrizo citrange”, “Forner-Alcaide 5”, “Forner-Alcaide 418”, and “Forner-Alcaide 517” being those that presented the highest concentration.

## 1. Introduction

Citrus is one of the main cultivated fruits worldwide [1]. Among the different citrus fruits (oranges, lemons, limes, grapefruit, and mandarins), the mandarin (*Citrus reticulata*) is gaining popularity due to its economic and nutritional value [2]. Mandarin production has reached 38 million tons in 2020 [3]. Currently, China is the largest producer of mandarins (23.12 mln. tons), followed by Spain (2.17 mln. tons), Turkey (1.58 mln. tons), and Brazil (1.02 mln. tons) [3]. Throughout the world, citrus fruits are one of the most important fruits, especially in juice production [4,5]. However, mandarins are mainly consumed fresh, although they have a shorter shelf life than other citrus fruits [6]. In this sense, Spain has had notable success with its seedless clementine varieties in Europe and the United States [7]. The main reasons for the fresh consumption of mandarins are that they are easy to peel; have a desirable flavor; and their content of vitamin C (≈25.8 mg/100 mL), flavonoids (≈38.97 mg rutin equivalent g^−1^ DW), and total phenolics (≈59.3 mg GAE/100 mL) [6,8]. The presence of citrus phenolic compounds contributes to the sensory quality of the fruit, in addition to being associated with the reduction of cardiovascular diseases and some types of cancer [9,10]. Moreover, aromas and volatile compounds play a fundamental role since they are responsible for the flavor of the fruit, so aromas are an important contributor to the sensory quality of these fruits and their derivatives [11,12,13].

On the other hand, farmers depend not only on the yield but also on the quality of the fruit [14]. In this sense, rootstocks play an important role since they help crops adapt to climate and soil conditions, as well as being a method of defense against climate change [9]. The selection of the appropriate rootstock is crucial for the success of the crop [15]. The identification of markers linked to citrus flavor and aroma can facilitate genetic improvement and the release of new superior varieties [8]. Some authors have shown that rootstocks affect the quality of citrus fruits, for example soluble solids content, acidity, ripening index, composition sugars and organic acids, antioxidant activity, and total phenolics, among others [14,16,17]. Currently, consumers demand higher-quality fruit that is produced sustainably [18]. Therefore, obtaining higher-quality citrus (internal and external) is essential. Then, the new studies carried out no longer focus exclusively on the yield and optimization of crops but instead choose to evaluate the effect of rootstocks on the quality of fruits [9]. Furthermore, there is little information on the effect of rootstock on volatile compounds in citrus.

For all the above-mentioned reasons, the objective of this study was to identify the influence of 9 generative rootstocks (“Carrizo citrange”, “Swingle citrumelo CPB 4475”, “Macrophylla”, “Volkameriana”, “Forner-Alcaide 5”, “Forner-Alcaide V17”, “C-35”, “Forner-Alcaide 418”, and “Forner-Alcaide 517”) on the volatile composition of “Clemenules” mandarin (*Citrus clementina* Hort. ex Tan.). This information can be used to improve the citrus market, which can provide sustainable economic opportunities for growers and be useful in promoting the use of rootstocks that induce greater citrus aroma.

## 2. Materials and Methods

### 2.1. Plant Material

“Carrizo citrange”, “Swingle citrumelo CPB 4475”, “Macrophylla”, “Volkameriana”, “C-35”, and four new hybrid selections, obtained in the rootstock breeding program carried out at IVIA (Instituto Valenciano de Investigaciones Agrarias) since 1974 (Table 1), were tested as rootstocks for “Clemenules” (selection virus-free INIASEL 22). Seeds of “Carrizo” citrange and “Cleopatra” mandarin were obtained from the germplasm collection of rootstocks at IVIA, and the seeds of the hybrids were obtained from the plants obtained in the citrus rootstock breeding program.

The trial was located in Museros, at ANECOOP’s “Masía del Doctor” (Valencia, Spain). The soil type of the trial plot as well as the fertilization applied were those described by Legua et al. [17].

### 2.2. Preparation of Juice

The mandarin “Clemenules” (*Citrus clementina* Hort. ex Tan.) fruits were harvested at optimum maturity (>12 °Brix). The juice preparation was carried out according to the methodology proposed by Legua et al. [17].

### 2.3. Volatile Composition

The determination of volatile compounds in the mandarin juice was carried out following the method described by Cano-Lamadrid et al. [19], using the headspace solid-phase micro-extraction (HS-SPME) method with slight modifications. A SPME 50/30 mm DVB/CAR/PDMS (Divinylbenzene/Carboxen/Polydimethylsiloxane) fiber (Supelco) was used for the extraction. The exposure time was 50 min at a temperature of 40 °C and with constant agitation (600 rpm). Then, desorption of the volatile compounds from the fiber was carried out in the injection port of the gas chromatograph for 3 min at 230 °C. Volatile compounds were analyzed and identified using a Shimadzu GC-17A gas chromatograph coupled to a Shimadzu QP-5050A mass spectrometer (Shimadzu Corporation, Kyoto, Japan). The analysis was carried out from 45 to 400 *m*/*z* with an electronic impact (EI) of 70 eV in 1 scan/s mode. The GC-MS system consisted of a TRACSIL Meta X5 column containing 95% dimethylpolysiloxane and 5% diphenylpolysiloxane (Teknokroma S. Co., Ltd., Barcelona, Spain; 30 m × 0.25 mm i.d., 0.25 µm film thickness). The oven program started at 80 °C with an increase of 3 °C/min from 80 °C to 210 °C and hold for 1 min. After this, an increase of 25 °C/min from 210 °C to 300 °C was maintained for 3 min. The injector and detector temperatures were 230 and 300 °C, respectively. Helium was used as the carrier gas (column flow rate of 0.6 mL/min).

Three methods were used to identify volatile compounds: (i) retention rates and their comparison with the literature; (ii) retention times of pure chemical compounds; (iii) mass spectra of authentic chemical compounds and the spectral library of the National Institute of Standards and Technology (NIST) database. Only fully identified compounds have been described. The analysis of the volatile composition was run in triplicate.

### 2.4. Statistical Analysis

To carry out the statistical analysis, the software XLSTAT (Addinsoft 2016.02.270444 version, Paris, France) was used. Two-way analysis of variance (ANOVA) and Tukey’s multiple range test were used to compare experimental data and determine significant differences between rootstocks (*p* < 0.05). Principal component analysis (PCA) using Pearson correlation was also run.

## 3. Results and Discussion

A total of 71 volatile compounds (Table 2) were identified in the analyzed samples. Limonene stands out among the 10 main compounds (Table 3), with an average of 7998.4 µg L^−1^, which was expected since it is the main volatile compound in citrus [20,21], followed by: myrcene (293.7 µg L^−1^), linalool (247.4 µg L^−1^), valencene (122.1 µg L^−1^), decanal (119.9 µg L^−1^), ethanol (106.4 µg L^−1^), ethyl butyrate (84.8 µg L^−1^), terpinen-4-ol (80.5 µg L^−1^), octanal (65.5 µg L^−1^), and 1-octanol (40.7 µg L^−1^). It is interesting to note that limonene and valencene may affect the perception of other volatiles [22,23].

Looking at the main compounds detected, the “Forner-Alcaide 517” rootstock obtained the highest values in ethanol (200 µg L^−1^). This volatile compound can accumulate in very high concentrations in mandarins due to the fermentation process caused by a lack of oxygen [12]. In addition, the rootstock “Forner-Alcaide 517” stood out together with “Carrizo citrange” for its high content in limonene (12,785 and 12,278 µg L^−1^, respectively), myrcene (485 and 521 µg L^−1^, respectively), linalool (323 and 416 µg L^−1^, respectively), octanal (171 and 149 µg L^−1^, respectively), and 1-octanol (63.3 and 59.0 µg L^−1^, respectively). Several authors include limonene, linalool, terpinene-4-ol (wood), and myrcene as key aroma volatile compounds in mandarin juice [8,12,19,26,27]. Furthermore, α-pinene is considered a positive contributor to citrus fruits aroma [22,28]. In this case, “Carrizo citrange” and “Forner-Alcaide 517” showed the highest values of this volatile compound (66.1 and 58.4 µg L^−1^, respectively), and “Macrophylla”, “Volkameriana”, and “C-35” the lowest (13.6, 14.4, and 15.0 µg L^−1^, respectively). Furthermore, “Carrizo citrange” had the highest values of decanal (308 µg L^−1^), valencene (215 µg L^−1^), and terpinen-4-ol (132 µg L^−1^). This last compound, in certain cases, terpinen-4-ol can be considered an unpleasant aroma in mandarin fruits [21,29]. On the other hand, Chen et al. [11] found only 26 volatile compounds present in juice mandarins, with limonene being the main compound (11,617.3 µg L^−1^), followed by γ-terpinene (961.6 µg L^−1^), *β*-myrcene (721.9 µg L^−1^), *α*-pinene (257.7 µg L^−1^), and *β*-pinene (122.0 µg L^−1^). The values of these volatile compounds were higher than those found in this study. In contrast, 114 volatile compounds were found by Bai et al. [22], who identified D-limonene, β-myrcene, and α-pinene as the main compounds in citrus peel oil, and 167 aroma volatiles were identified by Yu et al. [8] in mandarin juice, including ethanol, acetone, 2-methyl-2-propanol, *α*-pinene, myrcene, *α*-terpinene, *p*-cymene, limonene, terpinolene, and linalool, which are present in all citrus genotypes.

These results demonstrate that rootstocks significantly affect the volatile composition of citrus. Similar results were found by Aguilar-Hernández et al. [16] in lemon fruits. In the same way, Castle [14] showed that rootstocks have effects on the quality factors of citrus fruits. The rootstocks under study were also studied by Legua et al. [17], showing their influence on the composition of bioactive constituents in mandarins. Furthermore, Saini et al. [30] found that “Kinnow” mandarin juice grafted on “Pectinifera” had the highest levels of limonene and therefore the highest values of total volatile compounds, while the same mandarin grafted on “Shekwasha” had the highest levels of β-pinene, dodecylaldehyde, octanal, α-terpineol, terpinen-4-ol, peraldehyde, nonanal, isoleucine, linalool, and hexanal. Furthermore, Raddatz-Mota et al. [31] discovered that rootstocks not only affect the volatile profile but also have an effect on the presence or absence of certain volatile compounds in the fruit. This was the case for “Persian” lime, in which *β*-myrcene was only found in two of the five rootstocks studied, while the compounds *β*-thujene and dodecane were only found in the rootstocks “Volkamer” lemon and “C-35”.

Grouping the compounds by their chemical families (Figure 1), in general, the terpenes stand out over the rest of the chemical families, being the majority in the “Carrizo citrange” and “Forner-Alcaide 517” rootstocks (Table 4). In these same rootstocks, aldehydes and alcohols were also the majority. The esters presented a higher concentration in the samples of the rootstocks “Forner-Alcaide 5” and “Forner-Alcaide 517”, while “Forner-Alcaide 418”, “Macrophylla”, and “Volkameriana” had the lowest concentrations. During the ripening of mandarins, there is an increase in the concentration of esters, which are responsible for the fruity and sweet aroma, which can lead to unpleasant aromas or the perception that the fruit is over-ripe [24]. These results agree with those obtained by Morales-Alfaro et al. [9], Benjamin et al. [15], and Cano-Lamadrid et al. [19].

To gain a better understanding of the relationships established between the volatile compounds found (72), a principal component analysis (PCA) was performed on the experimental results (Figure 2). The PCA explained 68.61% of the variables, with the F1 axis being the one that explained most of the data (55.33%). The PCA showed that the rootstocks “Carrizo citrange”, “Forner-Alcaide 5”, “Forner-Alcaide 418”, and “Forner-Alcaide 517” were characterized by the most volatile compounds detected, with “Carrizo citrange” being the one that presented a different volatile profile from the other 3 rootstocks. These results agree with those obtained in the analysis of volatile compounds, in which it was these four rootstocks that presented a higher total concentration of volatile compounds. The rootstocks “Forner-Alcaide 517” and “Forner-Alcaide 5” have a common parent, so it was expected that they would present similar results [16,32].

## 4. Conclusions

The results obtained show that the rootstock used in the cultivation of mandarins affects the volatile content of its juice. In this case, the rootstocks that showed the highest volatile concentration were “Carrizo citrange”, “Forner-Alcaide 5”, “Forner-Alcaide 418” and “Forner-Alcaide 517”, while “Macrophylla”, “Volkameriana”, and “C-35” were the least. However, more research is needed to assess the effects of the environment and other factors on rootstocks and their effect on citrus juice properties.

## Figures and Tables

**Figure 1 foods-12-01599-f001:**
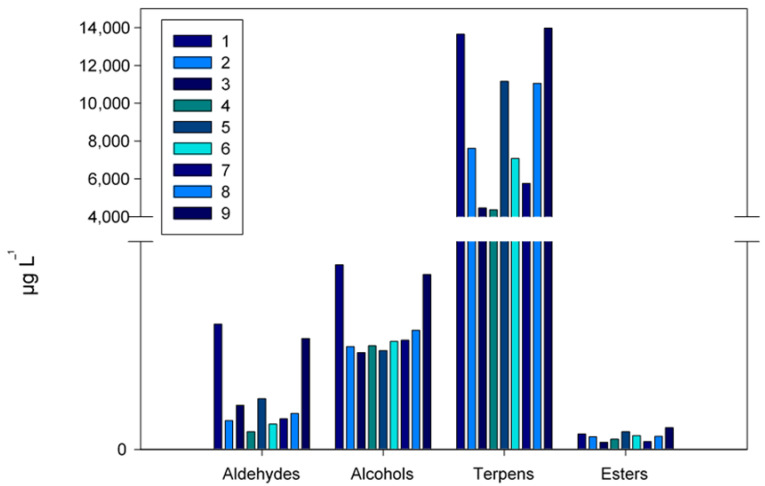
Concentration (µg L^−1^) of the main chemical families identified in mandarins (*Citrus clementina* Hort. ex Tan.). Rootstock: “Carrizo citrange” (1), “Swingle citrumelo CPB 4475” (2), “Macrophylla” (3), “Volkameriana” (4), “Forner-Alcaide 5” (5), “Forner-Alcaide V17” (6), “C-35” (7), “Forner-Alcaide 418” (8), and “Forner-Alcaide 517” (9).

**Figure 2 foods-12-01599-f002:**
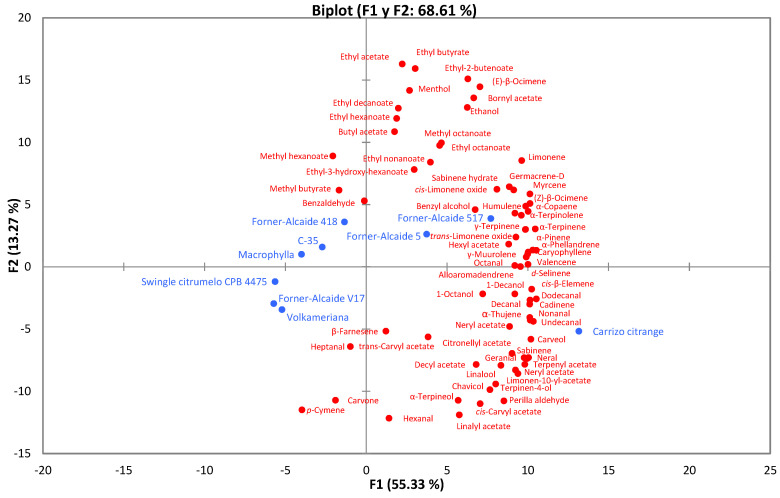
A principal component analysis (PCA) plot showing the relationships among volatile compounds and the factor rootstock.

**Table 1 foods-12-01599-t001:** Pedigree of the nine rootstocks tested for “Clemenules” mandarin.

	Rootstocks	Botanical Name
1	Carrizo citrange	*Citrus sinensis* (L.) Osb. × *Poncirus trifoliata* (L.) Raf.
2	Swingle citrumelo CPB 4475	*C. paradisi* × *P. trifoliata*
3	Macrophylla	*C. macrophylla* Wester
4	Volkameriana	*C. volkameriana* Ten. and Pasq.
5	Forner-Alcaide 5	*C. reshni* × *P. trifoliata*
6	Forner-Alcaide V17	*C. volkameriana* × *P. trifoliata*
7	C-35	*C. sinensis* × *P. trifoliata*
8	Forner-Alcaide 418	(*C. sinensis* x *P. trifoliata*) × *C. deliciosa* Ten.
9	Forner-Alcaide 517	*C. nobilis* Lour. × *P. trifoliata*

**Table 2 foods-12-01599-t002:** Retention indexes (RT), kovats indexes (KI EXP: kovats index experimental, and LIT: kovats index literature), and principal descriptors of the volatile compounds identified in “Clemenules” mandarin juice [16,24,25].

	Compound	RT	KI (Exp)	KI (Lit)	Descriptors
V1	Ethanol	5.14	498	482	Ethanol
V2	Ethyl acetate	5.63	613	608	Pleasant, fruity
V3	Methyl butyrate	6.35	694	719	Fruity, sweet
V4	Ethyl butyrate	7.28	797	799	Fruity, sweet
V5	Hexanal	7.36	803	801	Green, grassy
V6	Butyl acetate	7.48	809	813	Fruity
V7	Ethyl-2-butenoate	8.11	841	834	--
V8	Heptanal	9.42	906	902	Oily, fatty
V9	Methyl hexanoate	9.89	922	924	Fruity
V10	α-Thujene	10.18	933	933	Wood, green, herb
V11	α-Pinene	10.51	944	940	Pine, turpentine
V12	Benzaldehyde	11.54	981	970	Almond, cherry
V13	Sabinene	11.68	986	978	Pepper, turpentine, wood
V14	Myrcene	11.97	996	995	Musty, wet soil
V15	Ethyl hexanoate	12.17	1002	1000	Fruity, sweet, green
V16	Octanal	12.50	1011	1006	Citrus, green, herbal
V17	Hexyl acetate	12.63	1014	1011	Fruity, green, sweet
V18	α-Phellandrene	12.83	1019	1025	Citrus, herbal, green, woody
V19	d-3-Carene	12.95	1022	1013	Citrus, herbal, woody
V20	α-Terpinene	13.23	1029	1023	Lemony, citrus
V21	p-Cymene	13.55	1037	1027	Woody, spicy
V22	Limonene	13.92	1047	1039	Citrus, fresh
V23	Benzyl alcohol	14.01	1049	1040	Floral, fruity, sweet
V24	(Z)-β-Ocimene	14.10	1051	1050	Herbal, sweet
V25	(E)-β-Ocimene	14.53	1062	1053	Herbal, sweet
V26	γ-Terpinene	14.76	1068	1066	Lemony, citrus
V27	1-Octanol	15.08	1076	1072	Waxy, green, citrus, floral
V28	Sabinene hydrate	15.70	1092	1096	Herbal, minty, green
V29	α-Terpinolene	15.89	1097	1092	Citrus, pine
V30	Linalool	16.28	1106	1101	Floral, green, citrus, woody
V31	Nonanal	16.48	1111	1102	Pine, floral, citrus
V32	Methyl octanoate	17.12	1125	1127	Waxy, green, orange, herbal, sweet
V33	Ethyl-3-hydroxy-hexanoate	17.53	1134	1130	Fruity, woody, spicy, green
V34	*cis*-Limonene oxide	17.69	1138	1132	Fresh citrus
V35	*trans*-Limonene oxide	17.97	1144	1138	Fresh citrus
V36	Menthol	18.72	1161	1160	Minty
V37	Terpinen-4-ol	20.25	1195	1192	Peppery, woody, sweet, musty
V38	Ethyl octanoate	20.35	1197	1200	--
V39	α-Terpineol	20.86	1208	1192	Oil, anise, mint
V40	Decanal	20.98	1211	1216	Beefy, musty
V41	Carveol	21.73	1227	1220	Minty
V42	Chavicol	22.04	1234	1251	Herbal
V43	Neral	22.65	1247	1235	Lemon
V44	Linalyl acetate	22.88	1251	1250	Herbal, green, citrus, woody, floral
V45	Carvone	23.18	1258	1254	Spearmint, caraway
V46	Geranial	23.97	1275	1277	Lemon, mint, floral
V47	1-Decanol	24.23	1280	1274	Fatty, waxy, floral, citrus
V48	Perilla aldehyde	24.74	1291	1271	--
V49	Ethyl nonanoate	24.95	1296	1296	Fruity, rose, waxy, rum, wine
V50	Bornyl acetate	25.13	1300	1285	Woody, balsamic, pine, herbal
V51	Undecanal	25.70	1312	1307	Floral, citrus, green
V52	*cis*-Carvyl acetate	26.60	1332	1334	Minty, green, herbal
V53	*trans*-Carvyl acetate	26.91	1338	1341	Minty, green, herbal
V54	Citronellyl acetate	27.48	1351	1354	Floral, green, fuity, citrus, woody
V55	Terpenyl acetate	27.68	1355	1351	Herbal, citrus
V56	Neryl acetate	28.83	1380	1368	Fruity, floral, citrus
V57	α-Copaene	29.27	1390	1377	Woody, spicy, honey
V58	Ethyl decanoate	29.53	1395	1397	Waxy, fruity
V59	*cis*-β-Elemene	29.77	1400	1381	Herbal
V60	Decyl acetate	30.16	1409	1408	Waxy, soapy, citrus
V61	Dodecanal	30.36	1414	1409	Citrus, green, floral
V62	Limonen-10-yl-acetate	30.51	1417	na	Fruity
V63	β-Farnesene	30.99	1428	1431	Woody, citrus, herbal
V64	Caryophyllene	31.43	1437	1430	Spicy, Woody, clove
V65	Germacrene-D	31.80	1446	1449	Woody, spicy
V66	Alloaromadendrene	32.57	1463	1462	Woody
V67	Humulene	33.03	1473	1461	Woody, spicy-clove
V68	Valencene	34.61	1509	1496	Citrus, fruity, woody
V69	d-Selinene	34.81	1514	1496	--
V70	Cadinene	35.53	1531	1516	Fresh woody
V71	γ-Muurolene	35.88	1539	1530	Woody, herbal, spicy

**Table 3 foods-12-01599-t003:** Concentrations (µg L^−1^) of volatile compounds in “Clemenules” mandarin (*Citrus clementina* Hort. ex Tan.) juice.

		Carrizo Citrange	Swingle Citrum.	Macrophylla	Volkameriana	Forner-Alcaide 5	Forner-Alcaide V17	C-35	Forner-Alcaide 418	Forner-Alcaide 517
Compound	ANOVA ^Ϯ^	µg L^−1^								
Ethanol	***	116 bc^‡^	94.3 cd	67.5 de	76.8 de	88.9 cd	121 bc	46.9 e	146 b	200 a
Ethyl acetate	***	1.7 c	2.7 b	1.3 c	2.5 b	2.9 b	3.7 a	1.4 c	2.5 b	4.0 a
Methyl butyrate	***	3.1 e	4.8 cd	2.9 e	0.0 f	5.1 c	8.0 b	10.2 a	3.5 de	4.3 cde
Ethyl butyrate	***	72.1 b	105 a	64.2 b	59.1 b	105 a	114 a	30.3 c	108 a	106 a
Hexanal	***	10.6 b	7.2 c	7.3 c	14.3 a	7.2 c	5.7 cd	2.7 e	4.9 de	5.7 cd
Butyl acetate	***	21.1 cd	26.1 bc	13.2 e	16.9 de	25.6 bc	13.9 e	35.1 a	31.2 ab	31.6 ab
Ethyl-2-butenoate	***	3.3 cde	3.3 bcd	2.4 de	2.2 e	4.4 b	3.7 bc	1.0 f	4.3 bc	7.1 a
Heptanal	***	1.2 bc	0.9 cd	1.3 b	2.3 a	0.4 e	0.9 cd	0.6 de	1.4 b	1.4 b
Methyl hexanoate	***	1.2 cd	2.8 b	1.7 cd	1.9 c	2.6 b	3.9 a	1.2 d	1.5 cd	1.9 c
α-Thujene	***	2.2 a	0.3 d	0.3 d	0.4 d	0.7 c	0.4 d	0.3 d	0.8 c	1.2 b
α-Pinene	***	66.1 a	22.7 c	13.6 d	14.4 d	38.4 b	22.5 c	15.0 d	36.9 b	58.4 a
Benzaldehyde	***	1.3 cd	4.8 a	1.0 cde	0.7 de	0.5 e	1.1 cd	1.5 c	1.1 cd	2.9 b
Sabinene	***	24.4 a	3.9 cd	2.8 cd	2.5 cd	4.2 c	2.5 cd	1.3 d	4.0 c	11.8 b
Myrcene	***	521 a	228 cd	131 e	137 de	361 b	241 c	170 cde	369 b	485 a
Ethyl hexanoate	***	15.1 c	22.8 ab	8.9 d	14.4 c	19.0 bc	25.7 a	5.7 d	18.1 bc	17.6 c
Octanal	***	149 a	30.2 cd	54.9 b	5.6 e	52.6 bc	21.7 de	53.3 bc	48.8 bc	171 a
Hexyl acetate	***	12.7 ab	4.9 cd	3.2 d	6.4 c	14.8 a	6.3 c	3.3 d	3.4 d	11.2 b
α-Phellandrene	***	10.3 a	4.1 c	2.4 de	2.2 e	8.1 b	4.0 cd	3.9 cd	4.2 c	7.7 b
*d*-3-Carene	***	32.3 a	10.7 de	6.2 e	6.2 e	35.9 a	8.2 de	12.7 cd	18.0 c	25.0 b
α-Terpinene	***	14.7 a	7.0 c	3.6 e	3.3 e	9.9 b	4.6 cd	4.7 cd	10.0 b	10.6 b
*p*-Cymene	***	2.1 b	1.7 b	1.9 b	5.0 a	1.1 c	1.6 b	2.1 b	2.0 b	1.1 c
Limonene	***	12,278 ab	7004 c	4021 e	3879 e	10,124 b	6353 cd	5348 cd	10,194 b	12,785 a
Benzyl alcohol	***	69.0 a	35.7 b	17.4 cd	20.1 cd	11.0 d	37.2 b	26.0 bc	67.5 a	69.0 a
(Z)-β-Ocimene	***	34.1 a	14.6 d	7.2 ef	5.6 f	24.7 bc	12.7 de	10.5 def	22.4 c	28.4 ab
(E)-β-Ocimene	***	1.2 bc	0.5 c	0.2 d	0.2 d	1.6 b	1.2 bc	1.0 c	1.5 b	2.0 a
γ-Terpinene	***	43.7 a	19.0 c	11.6 d	10.3 f	33.6 b	14.5 cd	13.7 cd	29.6 b	30.0 b
1-Octanol	***	59.0 a	40.7 bc	29.8 cd	33.7 c	29.0 cd	39.6 bc	44.1 b	27.2 d	63.3 a
Sabinene hydrate	***	2.2 a	0.9 bc	0.6 c	0.5 c	2.3 a	1.0 bc	1.5 b	1.3 b	2.2 a
α-Terpinolene	***	22.3 a	8.6 c	5.2 d	3.8 e	20.8 a	7.8 cd	8.5 c	13.9 b	15.8 b
Linalool	***	416 a	196 c	177 c	235 bc	197 c	209 bc	274 ab	199 c	323 a
Nonanal	***	47.3 a	11.9	14.1 cd	10.0 d	17.9 c	10.6 d	11.5 cd	12.3 cd	38.1 b
Methyl octanoate	***	2.3 b	1.7 bc	1.2 c	2.4 b	4.0 a	2.8 b	1.1 c	2.3 b	2.4 b
Ethyl-3-hydroxy-hexanoate	***	8.7 bc	10.1 ab	6.6 c	12.7 a	12.7 a	11.6 a	2.8 d	8.4 bc	12.1 a
*cis*-Limonene oxide	***	3.4 a	1.6 de	2.0 cd	1.4 de	3.6 a	3.2 ab	1.0 e	2.6 bc	2.9 ab
*trans*-Limonene oxide	***	1.5 a	0.8 b	0.4 c	0.7 b	0.8 b	0.5 bc	0.5 bc	0.8 b	1.7 a
Menthol	***	0.8 c	1.1 c	1.2 c	0.9 c	1.6 b	1.0 c	1.4 bc	1.2 c	2.2 a
Terpinen-4-ol	***	132 a	64.5 c	105 b	64.2 c	76.4 c	53.9 c	56.c8	71.1 c	101 b
Ethyl octanoate	***	14.1 b	13.3 b	8.4 cd	11.3 bc	13.2 b	11.5 bc	6.4 d	18.5 a	13.0 b
α-Terpineol	***	33.1 a	21.4 d	30.1 b	24.9 bc	23.3 bc	21.8 cd	29.0 b	22.4 c	31.9 ab
Decanal	***	308 a	59.6 de	104 c	30.1 e	123 c	56.3 de	55.1 de	80.1 cd	263 b
Carveol	***	6.1 a	2.0 d	2.1 d	1.7 d	3.1 bc	2.0 d	2.2 cd	2.4 cd	3.7 b
Chavicol	***	40.2 a	28.6 b	26.2 b	32.7 ab	32.5 ab	25.7 b	32.1 ab	25.9 b	32.8 ab
Neral	***	15.3 a	3.0 de	3.9 de	3.6 de	6.1 bc	3.7 de	2.9 e	4.8 cd	7.5 b
Linalyl acetate	***	8.6 a	4.9 b	5.1 b	2.5 de	2.4 de	4.8 bc	1.9 e	2.3 e	3.6 cd
Carvone	***	14.3 b	9.8 cd	12.9 bc	33.8 a	12.2 bc	10.1 bcd	10.2 bcd	7.6 d	11.1 bcd
Geranial	***	25.4 a	4.9 c	5.7 c	4.7 c	10.9 b	4.4 c	4.5 c	6.0 c	12.2 b
1-Decanol	***	16.0 a	9.3 c	8.9 c	7.9 c	12.8 ab	8.0 c	12.2 bc	8.9 c	14.1 ab
Perilla aldehyde	***	22.3 a	11.4 bc	14.1 b	10.1 cd	12.9 bc	12.4 bc	10.1 cd	7.6 d	15.0 b
Ethyl nonanoate	***	4.2 c	3.4 c	3.0 c	3.3 c	2.8 c	1.8 c	11.1 b	3.8 c	33.2 a
Bornyl acetate	***	2.4 bc	2.1 bc	1.5 c	1.4 c	2.2 bc	2.1 bc	2.0 bc	2.8 b	4.2 a
Undecanal	***	8.6 a	2.2 c	2.6 c	2.1 c	5.1 b	2.0 c	1.9 c	2.0 c	6.0 b
*cis*-Carvyl acetate	***	10.2 a	2.4 cd	5.0 b	3.5 c	2.7 c	6.4 b	1.3 d	1.2 d	5.0 b
*trans*-Carvyl acetate	***	6.3 a	5.5 a	5.2 a	1.9 cd	3.1 bc	5.9 a	1.5 d	1.8 cd	3.8 b
Citronellyl acetate	***	7.6 a	2.7 cd	2.3 cd	2.2 d	4.2 b	3.0 cd	2.4 cd	3.4 bc	2.9 cd
Terpenyl acetate	***	40.1 a	10.8 c	10.7 c	11.9 c	21.6 b	10.9 c	12.4 c	8.1 c	21.6 b
Neryl acetate	***	39.5 a	13.6 c	9.8 d	14.2 c	25.0 b	13.0 c	12.8 c	16.4 c	16.0 c
α-Copaene	***	2.1 a	0.6 c	0.7 bc	0.5 c	2.0 a	1.1 b	0.5 c	0.8 bc	2.4 a
Ethyl decanoate	***	3.1 c	3.5 bc	1.6 e	1.8 de	4.5 b	2.6 cd	2.0 de	7.0 a	2.6 cd
*cis*-β-Elemene	***	9.1 a	2.7 c	2.6 c	3.0 c	5.5 b	4.5 b	2.6 c	4.6 b	5.7 b
Decyl acetate	***	7.9 a	4.5 bc	4.4 bc	1.6 ef	3.4 cd	4.8 b	1.5 de	2.7 f	3.3 cd
Dodecanal	***	13.2 a	2.6 de	3.4 de	1.8 e	6.2 c	2.6 de	2.3 de	3.8 d	10.0 b
Limonen-10-yl-acetate	***	33.5 a	7.3 e	10.0 de	8.9 de	19.7 b	8.2 cd	12.3 de	6.8 e	16.1 bc
β-Farnesene	***	17.5 b	15.7 b	17.9 b	5.2 d	10.7 c	22.0 a	6.3 d	7.8 cd	7.6 cd
Caryophyllene	***	8.8 a	1.9 de	2.3 de	2.1 de	5.3 c	2.8 d	1.2 e	4.4 c	7.4 b
Germacrene-D	***	2.0 b	0.7 c	0.6 c	0.7 c	1.6 bc	1.0 c	0.7 c	0.8 c	3.1 a
Alloaromadendrene	***	4.7 a	1.1 ef	0.8 ef	2.1 cd	4.0 ab	2.4 c	0.5 f	1.5 de	3.3 b
Humulene	***	2.0 a	1.0 b	1.0 b	0.8 b	1.9 a	1.1 b	1.1 b	1.1 b	2.0 a
Valencene	***	251 a	47.8 e	60.1 de	93.4 cd	200 b	124 c	21.1 f	101 cd	201 b
d-Selinene	***	25.8 a	5.8 d	6.6 d	10.2 c	20.3 b	12.9 c	2.5 e	10.6 c	19.6 b
Cadinene	***	5.0 a	1.1 cd	1.6 c	1.6 c	2.5 b	1.6 c	0.5 d	1.1 cd	4.3 a
γ-Muurolene	***	14.5 a	2.7 e	3.5 de	5.7 cd	11.6 b	6.8 c	1.2 e	6.0 c	11.5 b
**TOTAL**	***	15,225 a	8309 bc	5170 d	4999 d	11,968 ab	7777 bc	6473 cd	11,861 ab	15,446 a

^Ϯ^ *** significant at *p* < 0.001. ^‡^ Values (mean of 3 replications) followed by the same letter within the same volatile compound, were not significantly different (*p* < 0.05), according to Tukey’s least significant difference test. Rootstock: “Carrizo citrange”, “Swingle citrumelo CPB 4475”, “Macrophylla”, “Volkameriana”, “Forner-Alcaide 5”, “Forner-Alcaide V17”, “C-35”, “Forner-Alcaide 418”, and “Forner-Alcaide 517”.

**Table 4 foods-12-01599-t004:** Statistical differences found between the different chemical families. ^Ϯ^ *** significant at *p* < 0.001. Values followed by the same letter, within the same chemical family were not significantly different (*p* < 0.05), according to Tukey’s least significant difference test. Rootstock: “Carrizo citrange” (1), “Swingle citrumelo CPB 4475” (2), “Macrophylla” (3), “Volkameriana” (4), “Forner-Alcaide 5” (5), “Forner-Alcaide V17” (6), “C-35” (7), “Forner-Alcaide 418” (8), and “Forner-Alcaide 517” (9).

	1	2	3	4	5	6	7	8	9
ANOVA ^Ϯ^	***	***	***	***	***	***	***	***	***
Aldehydes	a	cd	bc	d	b	d	cd	bcd	a
Alcohols	a	b	b	b	b	b	b	b	a
Terpenes	ab	d	e	e	c	de	de	c	a
Esters	bc	cd	e	de	ab	cd	e	cd	a

## Data Availability

Data is contained within the article.

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
