# Peer review of "Effect of Rootstock on the Volatile Profile of Mandarins"

_foods, 2023, doi:10.3390/foods12081599_

Round 1

Reviewer 1 Report

Dear authors please consider the minor suggestions when revising your paper:

Line 63: influence of 9 generative rootstocks

Line 66: on the volatile composition of mandarin (Citrus clementina Hort. ex Tan.), which cultivar? 

Results from table 4 could be indicated in the figure 1. 

Author Response

Line 63: influence of 9 generative rootstocks

It has been added.

Line 66: on the volatile composition of mandarin (Citrus clementina Hort. ex Tan.), which cultivar?

It has been added. It can also be seen in the material and methods section line 73.

Reviewer 2 Report

This manuscript describes the importance of mandarin production and by using 9 different rootstocks, the authors identified different volatile compositions. Limonene is the main compound and terpens are the main chemical family.

Many literatures have proved the importance on rootstocks and also mentioned the volatile composition. The study is lack of novelty and the data analysis is inappropriate.

Two-way ANOVA should have the interaction between two factors, however, on the study, it has not been mentioned at all.

Also, the experiment is only tested on one location, unfortunately, G*E interaction often affects the final results, which mean the conclusion might be not accurate.

This study should enforce the statistical analysis part to dig more information in order to prove its novelty. 

Author Response

Many literatures have proved the importance on rootstocks and also mentioned the volatile composition. The study is lack of novelty and the data analysis is inappropriate.

There are numerous studies in which it has been studied that the rootstock has a decisive influence on the quality of the fruit. However, there are no studies that have seen the composition of volatile compounds as a function of the rootstock. Based on this, the most interesting rootstocks can be selected to obtain mandarins with a good aroma.

Two-way ANOVA should have the interaction between two factors, however, on the study, it has not been mentioned at all.

The objective of the study is not so much the comparison of factors but to know the profile of volatile compounds that each rootstock can induce in the fruit.

Also, the experiment is only tested on one location, unfortunately, G*E interaction often affects the final results, which mean the conclusion might be not accurate.

This factor has not been studied.

This study should enforce the statistical analysis part to dig more information in order to prove its novelty.

The novelty lies in the fact that these studies will allow farmers and the sector in general to select rootstocks based on whether they induce a better composition of volatile compounds.

Reviewer 3 Report

Manuscript with title: Effect of rootstock on the volatile profile of mandarins is very interesting and well written.

 Presented manuscript is on high scientific level. The manuscript authors the introduction section presented the current state of knowledge on the experimental design. The topic is a new, as well present a very important aspects of

 The Abstract. Authors give a short presentation of manuscript. This part of manuscript is well constructed.  Please add main values for limonene content in selected examined samples.

 Keywords: please arrange the keywords in alphabetical order.

 The Introduction section includes all necessary information about examined objects and problems. The main goal of experiment was very well formulated.

Page 1, line 34: pleas change this huge numbers for shorter together with changing of units for example: not  (23,120,000 tons), only 23.12 mln. tonnes

 Page 1, line 43: please add values following listed bioactive compounds, for example:

“…. the content of vitamin C (value:….), flavonoids (value:….) and carotenoids (value:…), according to literature information.

 Material and method section is well described. I can see any mistakes in this part of manuscript.

 All Tables and Figures are well presented with good resolution, are easily for reading and understanding.

 The discussion section presents a good comparison of the obtained results with other results available in the data basis. I am impressed by the amount of results obtained and described by the authors. Very accurate and well-chosen literature for discussion and comparison.

Conclusion section is well formulated and corresponding with the obtained results.

 General opinion:  After carefully manuscript reading, I think, that presented experiment is a valuable. 

Author Response

Manuscript with title: Effect of rootstock on the volatile profile of mandarins is very interesting and well written.

Presented manuscript is on high scientific level. The manuscript authors the introduction section presented the current state of knowledge on the experimental design. The topic is a new, as well present a very important aspects of

 The Abstract. Authors give a short presentation of manuscript. This part of manuscript is well constructed. Please add main values for limonene content in selected examined samples.

It has been added as suggested.

Keywords: please arrange the keywords in alphabetical order.

It has been done as suggested.

 The Introduction section includes all necessary information about examined objects and problems. The main goal of experiment was very well formulated.

Page 1, line 34: pleas change this huge numbers for shorter together with changing of units for example: not  (23,120,000 tons), only 23.12 mln. tonnes

It has been changed.

Page 1, line 43: please add values following listed bioactive compounds, for example:

“…. the content of vitamin C (value:….), flavonoids (value:….) and carotenoids (value:…), according to literature information.

It has been added as suggested.

Material and method section is well described. I can see any mistakes in this part of manuscript.

It has been revised.

All Tables and Figures are well presented with good resolution, are easily for reading and understanding.

The discussion section presents a good comparison of the obtained results with other results available in the data basis. I am impressed by the amount of results obtained and described by the authors. Very accurate and well-chosen literature for discussion and comparison.

Conclusion section is well formulated and corresponding with the obtained results.

General opinion:  After carefully manuscript reading, I think, that presented experiment is a valuable.
